# The Influence of Forestry Management on the Selection of a Non-Vegetative Diet by the Eurasian Beaver (*Castor fiber* L.)

**DOI:** 10.3390/ani12212949

**Published:** 2022-10-26

**Authors:** Ondřej Mikulka, Petr Pyszko, Vlastimil Skoták, Jiří Kamler, Jakub Drimaj, Radim Plhal, Miloslav Homolka

**Affiliations:** 1Department of Forest Protection and Wildlife Management, Mendel University, 61300 Brno, Czech Republic; 2Department of Biology and Ecology, University of Ostrava, 70103 Ostrava, Czech Republic; 3Department of Game Management, Forestry and Game Management Research Institute, 15600 Praha, Czech Republic; 4Institute of Vertebrate Biology, Czech Academy of Sciences, 61300 Brno, Czech Republic

**Keywords:** behavior, managed forests, diet, damage, foraging, beaver management

## Abstract

**Simple Summary:**

In the non-growing season, beavers fell woody plants that they primarily use as a food source. It is well known that parameters such as woody plant availability, trunk diameter, and distance of woody plants from water influence food selection. Less information is available on the felling of specific species in different stand types where different forest management takes place. This makes the beaver a conflict species that causes damage to commercial forests, and information on foraging behavior in relation to environmental factors is very important. Especially as beaver populations continue to increase. In the three types of forest stands (monoculture, mixed forest, and natural forest), species selection, the intensity of felling, the distance of felled woody plants, and trunk diameter vary. The beaver, which generally prefers non-commercial tree species (willow, poplar, etc.), also adapts to and uses commercial oaks in commercial forests. These are most preferred at stand ages up to twenty years, i.e., smaller diameters but greater stand densities. The results suggest that by using knowledge of foraging behavior, economically profitable species can be protected by encouraging the non-commercial species most preferred by beaver (predominantly willow) near the shoreline. In addition, the natural tree composition around rivers or lakes will be promoted.

**Abstract:**

Native ecosystems have been transformed by humans into cultural landscapes, resulting in the disruption of natural interactions, with some species unable to adapt and disappearing from such landscapes. Other species were able to adapt their behavior to current environmental conditions. In some places, forest management has gradually transformed native diversified forests into stands converted for the greatest profit in the wood matter, thereby affecting the food availability for herbivores, among them the Eurasian beaver (*Castor fiber* Linnaeus, 1758). This study summarizes knowledge on non-vegetation diet selection by beavers in commercial and natural forests in the Czech Republic. A sample size of 25,723 woody plant specimens checked in 288 forest stands showed that beavers prefer willow, poplar, or hazel, but where these are less available, commercial tree species such as oak may dominate the diet and cause economic losses in forestry. Significant differences were also observed in a preference for different trunk diameters and distances from water in different types of stands. In terms of stand types, commercial monocultures are the most felled, while the probability of felling decreases with the age of stands. Based on these results and discussion, it is suggested that commercial forest stands including economically valuable woody plant species (e.g., oak, ash) could be protected through appropriate management measures, such as increasing the proportion of deciduous softwood stands along the riverbanks, which would distract the beavers from commercial woody plant species.

## 1. Introduction

Aside from a few small reservoir populations, the Eurasian beaver (*Castor fiber* Lin-naeus, 1758) was finally exterminated across much of their range in Europe in the 18th to 19th centuries [1]. Since then, its protected status has allowed it to spread (or be introduced) to all European countries [2,3]. The return of the beaver in the Czech Republic dates back to approximately 1990, since then the species has spread over a substantial area (ca. 30%) of the country [4], colonizing a range of habitat types.

Between its extinction and the return of a viable population, there have been significant changes in the landscape managed by human activity. This has led to both changes in beaver behavior in the cultural landscape, and the associated increasing conflict situations, including economic losses along regulated rivers related to waterways, hydroelectric power, sluices, and flood dams [5] and damage to agricultural fields [6] and forestry [7] related to beaver feeding behavior. Beaver presence can also bring significant positives to the agricultural landscape, including improved landscape water retention, renaturalization of streams [8], and increased biodiversity [9]. Thanks to these positive effects on the landscape, further support of populations through reintroduction has recently been encouraged in the UK [10].

In terms of feeding behavior, the beaver is an exclusive herbivore with a wide diet spectrum that follows two phases during the year. In the period from spring to autumn, the beaver feeds mainly on green biomass available along the banks of watercourses [11]. In winter, the beaver uses its extraordinary ability to cut down woody plants and the bast and young bark serve as its main source of winter food [12,13]. In Central Europe, the beaver mostly consumes woody plants from the beginning of October to the beginning of the growing season (March), with the most intensive felling of woody plants taking place between November and December. Beavers will then bite off branches from the fallen woody plants and use them to build an underwater cache for the period when the water’s surface is frozen, allowing them to survive the winter period [14,15,16].

Both Eurasian and North American (*Castor canadensis* Kuhl, 1820) beavers utilize an opportunistic foraging strategy [17,18] and may feed on many different woody plant species [19], depending on what is available in their immediate area. In some regions, for example, they commonly consume birch (*Betula* L. spp.) [13,20] or alder (*Alnus* L. spp.) [17], while at sites with a more variegated range of species they may show a preference for willow (*Salix* L. spp.) and poplar (*Populus* L. spp.) [11,17,18,21,22,23]. Beavers in the Czech Republic tend to show a clear preference for willows and poplars, with both species comprising up to 80% of the diet (by volume) in a recent wide-ranging study [24]. The same preference has also been confirmed in other studies [25,26,27].

However, most studies focus on a summary evaluation of one type of landscape or forests without forest management. At the same time, beaver behavior can vary significantly in different types of forest stands. Of particular importance as regards economic losses caused by beavers [7,28], it is important to know how they behave in different types of forest stands and if they prefer commercially significant woody plant species, particularly where their preferred species (willow, poplar) are less available. It is also important to answer the question of whether beavers can occur in an environment where there is a minimum of poplars and willows because some studies [29] state their presence in the environment as the main factor for the stability of the beaver population. Other factors that might also influence their preference for particular woody plants include the diameter of the woody plant being felled [17,30] and its distance from the water [31,32]. However, it is important to determine whether these factors have the same effect on all woody plants and different species composition (stands) or in some cases have no effect. Information on food selection in different forest stand types can answer important questions on how to adjust beaver management measures to reduce financial losses, while at the same time maintaining a viable beaver population.

In the Czech Republic, beavers most frequently colonize floodplain forests along lowland rivers [33]. Such forests tend to have a greater proportion of natural or near-natural stands with a high representation of non-commercial woody plant species, and most are easily accessible to beavers thanks to the river arms, backwaters, and wetlands that typify such floodplain areas. In commercial stands, however, which consist primarily of oak (*Quercus* L. spp.) and ash (*Fraxinus* L. spp.) [25], the natural species composition, including willow and poplar, is suppressed in favor of these commercial species (Figure A1). In both types of forest (commercial and natural), wide variability in these three factors (species composition, trunk diameter, and distance from water) can make predictions of damage based on ‘known’ species preferences difficult, with findings differing considerably from hitherto published data.

The goals of the paper, therefore, are:(a)To analyze on the individual woody plant level if there are preferences for some taxa of woody plants, especially for commercial species, i.e., *Quercus* and *Fraxinus*;(b)To analyze on the individual level, what is the role of distance from water and diameter of the woody plant on the probability of the felling by beaver and if these effects are taxa-specific;(c)To analyze on the patch level if some patch characteristics, primarily the type of forest stand (commercial monoculture, commercial mixed, natural), affect the proportion of felled basal area, the proportion of used woody plant taxa from their diversity in the stand, and the maximal distance of felled woody plants.

## 2. Materials and Methods

### 2.1. Study Sites

Data were collected from two localities on the River Morava in the Czech Republic, one denoted as River Morava north (200–280 m a.s.l.; “49.693° N, 17.131° E”) and the other River Morava south (150–165 m a.s.l.; “48.764° N, 17.032° E”). Both localities are covered with lowland alluvial forests, a substantial part of which consists of commercial stands. At the northern locality, natural and close-to-natural stands predominate, with oak and ash grown commercially in small monocultures of up to 0.5 ha. Generally speaking, the dominant woody plant species in non-commercial plots are European ash (*Fraxinus excelsior* L.), oak, elm (*Ulmus* L. spp.), and/or maple (*Acer* L. spp.) At the southern locality, the proportion of natural/near-natural growth is relatively low, while commercial species (dominated by oak) are grown in large monocultures of 1 to 2 ha. Woody vegetation tends to be dominated by oak, with a relatively high proportion of narrow-leaved ash (*Fraxinus angustifolia* Vahl.) and field maple (*Acer campestre* L.)

### 2.2. Data Collection

Data were collected from a total of 56 confirmed beaver territories during April and May of 2017 and 2018, with each territory including one feeding center. These feeding centers were detected by monitoring beaver signs near all rivers, ponds, and lakes within both study sites. Beaver signs were recorded by GPS and territories and feeding centers were subsequently evaluated in the Quantum GIS software package version 2.14 [34].

For each territory, four to six 5 × 20 m patches (total 288; number varied according to the size of feeding center) were staked out (short side to water). Each patch was then subdivided into four 5 × 5 m quadrats (Figure 1). If a beaver felled a woody plant at a distance > 20 m from the bank, the survey patch was extended such that it included all felled woody plants. In all cases, woody plants were classed as ‘felled’ if there were clear signs that they had been brought down by beaver activity and subsequently browsed; woody plants with small areas of browsed bark (up to one-third of the trunk circumference) were not included in the analysis, because it is not clearly determined whether this is food selection sign or sign of other types of behavior (e.g., possible territory marking). For each patch, all woody plants with a minimum trunk diameter of 0.5 cm (felled and non-felled) were recorded, including woody plants felled the previous winter (October–February). Next, the number and genus/species of all woody plants in each 5 × 5 m quadrat was recorded. As beavers tend to browse on smaller woody plants and drag them away (e.g., to a food cache), the trunk diameter of all woody plants was measured at 20 cm above the ground instead of measuring diameter at breast height (DBH). Each patch was designated as either commercial monoculture (N = 58), commercial mixed (N = 144), or near-natural/natural (N = 86), with the approximate age of all commercial woody plant species being noted. The “commercial” category was taken to include all patches mentioned in the forest management plan [35,36]. These were then further classified as either mixed stands or monocultures, with mixed stands having <90% commercial species and monocultures having >90% (Figure A1). Patch age was based on the classification list in the forest management plan.

### 2.3. Data Processing and Evaluation

The relative proportion of felled woody plants (proportion of felled woody plants to proportion of available woody plants, expressed as absolute basal area (m^2^)) was used to evaluate any differences in the proportion of felled woody plants, calculated using the formula: $BA=\frac{\pi d^{2}}{4}$, where *d* = trunk diameter.

Data were analyzed in R 4.2.1 [37]. First, we focused on general taxonomic differences in the proportion of felled woody plants. For this part of the analysis, we selected only woody plants with a distance ≤ 20 m from the water, as data from larger distances were obtained only for part of the patches and would bias the results. We worked to the genus level, as for many woody plants it was not possible to determine to species level. We included 17 woody plant genera, each with >60 individuals. We used generalized linear mixed models with multivariate normal random effects, using penalized quasi-likelihood (GLMMPQL) with binomial distribution and logistic link from the library “MASS” [38]. In the model, the presence/absence of damage on the woody plant was the response variable, the distance from water and diameter were used as covariates, a genus of the woody plant and its interaction with distance and trunk diameter were used as the explanatory variables, and patch nested in territory and locality was used as a random term. The final structure of the model was determined by stepwise selection based on Akaike Information Criterion (AIC). To evaluate the differences among genera in the proportion of felled woody plants, we chose Quercus as a control, as Quercus is a very important and frequent commercial woody plant belonging to the beavers’ preferred diet [30].

Then we focused on the patch level and asked, which patch characteristics determine the overall proportion of felled basal area, the proportion of used woody plant taxa from their diversity in the patch, and the maximal distance of felled woody plant in the patch. As potential explanatory variables, we used the overall density of woody plants (measured as a number of individuals per 25 m^2^), the density of commercial woody plants, the diversity of woody plant taxa, the proportion of commercial woody plants from all woody plants, approximal age of the patch, and especially type of the patch (monoculture, mixed or natural stand). The proportion of felled basal area was analyzed by GLMM with beta distribution and logistic link from the library “glmmTMB” [39]. The proportion of the used woody plant taxa from their diversity in the patch was analyzed by GLMM with binomial distribution and logistic link from library “lme4” [40]. The maximal distance of the felled woody plant in the patch was analyzed by GLMM with the negative binomial distribution.

The maximum number of function evaluations in GLMMs was set at 10^5^ and a less exact form of parameter estimation for the Gauss–Hermite approximation to the log-likelihood was used to solve algorithm convergence problems. In all mixed models working on patch level, locality nested in the territory was used as a random term. All these models were also weighted by the log of the number of woody plants in the patch and built based on AIC. Analysis of variance of all the models was performed using the library “car” [41]. The collinearity of factors was checked using variance inflation factor VIF < 2.5. All patches were created using libraries “jtools” [42] and “interactions” [43].

## 3. Results

### 3.1. Differences in Proportion of Felled Woody Plants

From the total number of 25,723 woody plant trunks, 46 species (29 genera) recorded in the northern and southern locations of floodplain forests were determined. In terms of the basal area, oak and ash dominated with 37.63% and 13.63%, respectively, followed by willow with 9.33% and poplar with 9.14% (Table 1). Using such a quantity of data allowed us to evaluate the level of patches and individuals of woody plants and to determine differences in the proportion of felled woody plants for 17 genera.

After controlling for the effect of spatial variability, distance from the water, and diameter, the probability of beaver felling on the individual woody plant taxa significantly differed (*d*f = 20,561, χ^2^ = 1163.79, *p* < 0.001, Figure 2). Compared to *Quercus*, one of the most frequently felled and simultaneously commercial woody plants used as control, *Alnus*, *Betula*, *Corylus*, and *Salix* were not significantly different from *Quercus* and thus also frequently felled in the diet. *Populus* was even significantly more frequently felled than *Quercus*. *Acer*, *Carpinus*, and *Ulmus* belong to the group of genera significantly less frequently felled than *Quercus*, but still more often eaten than the rest of the woody plant genera (Table 2).

### 3.2. Woody Plant Diameter and Distance of Felled Woody Plants from the Water

From the covariates, with the increasing distance from the water, the probability of beaver felling on woody plants decreased (*d*f = 20,561, χ^2^ = 642.74, *p* < 0.001, Figure 3a) but with significant interaction with the woody plant genus (*d*f = 20,561, χ^2^ = 251.35, *p* < 0.001, Figure 3b–d).

Diameter as the second covariate has also a significant effect on the probability of a beaver felling on woody plants. With increasing diameter, the probability of beaver felling decreased (*d*f = 20,561, χ^2^ = 115.45, *p* < 0.001, Figure 4a) but with significant interaction with the woody plant genus (*d*f = 20,561, χ^2^ = 340.30, *p* < 0.001, Figure 4b–d), e.g., with increasing diameter, the probability of felling did not decrease for *Quercus*, *Corylus* or *Sambucus*. 

### 3.3. Differences among Forest Types in Foraging Behavior

Working on the level of patches, the proportion of felled basal area decreased with the age of the patch (*d*f = 280, χ^2^ = 43.90, *p* < 0.001) and differed among type of forest patches (*d*f = 280, χ^2^ = 31.70, *p* < 0.001, Figure 5). Surprisingly, the natural type did not differ from monoculture patches (z = −0.98, *p* = 0.325), but the mixed type differed (z = −3.85, *p* < 0.001).

The proportion of woody plant taxa that showed herbivory from the patch diversity can be best explained by age of the patch and the density of woody plants. With increasing age of woody plants, this proportion decreased (*d*f = 279, χ^2^ = 34.66, *p* < 0.001, Figure 6a) but with the density of woody plants increased (*d*f = 279, χ^2^ = 78.85, *p* < 0.001, Figure 6b). The types of patches did not differ (*d*f = 279, χ^2^ = 2.71, *p* = 0.260, Figure 6c), with mixed (z = −1.56, *p* = 0.120) and natural (z = −1.44, *p* = 0.150) having similar proportion of consumed diversity as commercial ones.

The maximal distance of beaver felling can be best explained by age of the patch and the density of commercial woody plants. With increasing age of woody plants, this maximum distance decreased (*d*f = 279, χ^2^ = 32.50, *p* < 0.001, Figure 6d) but with the density of commercial woody plants increased (*d*f = 279, χ^2^ = 13.10, *p* < 0.001, Figure 6e). The types of forests differed (*d*f = 279, χ^2^ = 13.00, *p* = 0.002, Figure 6f), with nearly significantly larger maximal distances in mixed forests (z = 1.67, *p* = 0.094) and non-significantly shorter maximal distances in natural forests (z = −1.45, *p* = 0.147) than in the monoculture forest stands.

## 4. Discussion

Throughout their European and North American ranges, the diet of beavers is significantly affected by the woody species available in different areas, though poplars and willows are the most frequently reported species in the non-vegetation diet in most regions [17,18]. Indeed, it has even been suggested that beavers cannot permanently settle in a region where these species are absent [29,44]. This preference for willow and poplar is due to the species’ digestibility [45], their nutrient content, which is higher than other woody plant species over the winter period [18], or also because the bark is easier to separate from the wood [46].

Several previous studies have reported these same species as dominant in the diet of beavers inhabiting the Czech Republic [24,26], and our results from the northern Morava river basin correspond with these findings. In the southern Morava river basin, however, the predominant species in the beaver diet were oak and ash, with poplars and willows taken at lower levels. This was most likely due to the relatively low proportion of these species in the local environment. Nevertheless, beavers appear to thrive in the southern river basin floodplain forest, despite this switch to ‘lower quality’ species in the diet, the region has supported a relatively dense population for several decades [33].

In commercial monocultures, commercial tree species were in availability, while the proportion of preferred willows or poplars was zero. Despite this, the beaver was predominantly felling oak and it is clearly a suitable tree species. The selection of these woody plants is due to low stand diversity, suppression of non-commercial woody plants by forest management, and subsequent ease of access. Our results also suggest that the uniform diameter of woody plants in young monocultures may also play a role as beavers were able to access numerous woody plants of the preferred diameter without spending excessive energy in searching and felling [47].

In addition to tree genus, plant species selection was also influenced by the distance of suitable woody plants from water and trunk diameter. Our results showed that the probability of felling was greatest for the small diameters of most genera. This may be because a greater number of dietary categories were available at smaller diameters, including young trees, shrubs, saplings, and shoots from mature trees. On the other hand, due to their short-term exposure (e.g., saplings), relatively low biomass, and low density in some stands, trees with smaller diameters were sometimes “avoided”, probably due to the greater energy expenditure required to obtain them in such stands, and larger diameters were preferred, as in the case of oak or hazel. In addition, woody plant shoots that have been repeatedly felled may contain relatively high amounts of toxic secondary substances [48,49,50]. Finally, beavers have no natural predators at these sites, which generally affects beaver foraging [51,52], hence they can afford to expend more energy and time felling larger woody plants to obtain the greater biomass available from the crowns of mature woody plants. Previous studies have also shown that woody plant diameter preferences may change seasonally, with beavers having to gather large amounts of biomass over a short period around autumn for the construction of dams and winter food caches [16,53]. In such cases, the felling of mature and/or larger-diameter woody plants may be more effective as they can consume smaller-diameter woody plants on the spot towards the end of summer, during mild winter periods, and during the growing season.

Previous studies on the winter diet of beavers in the Czech Republic have provided similar results concerning the preference for plants with a smaller diameter. Authors [26] noted that most woody plants felled by beavers fell in the 2.6–6 cm (35% felled woody plants) diameter category and that fewer woody plants were felled in the 6.1–12 cm (27%) category, while other studies [25] found that woody plants of 1–10 cm (51%) were most frequently felled. Similar results have also been obtained outside the Czech Republic. Authors [17] noted that the majority of woody plants felled had diameters ranging from 1–5 cm, and other studies found that beavers preferred woody plants of 2–3 cm diameter [30]. In commercial forests in Poland, however, authors found that beavers preferred woody plant species of 21–30 cm diameter [54]. In each case, the differences in preference could be put down to variations in the relative abundance of species and age groups on offer. 

Beavers usually prefer the closest woody plants to the water. Most of the differences in each case could be attributed to local differences in the woody plants’ distribution from water. Using an aggregate sample, we were able to demonstrate that the probability of a woody plant being felled decreased with distance from water, with only exceptional cases far from a river attributable to the range of woody species on offer and the age structure of the stand. Overall, beavers appeared to concentrate their activity over a strip 1–10 m from the bank (58% felled woody plants basal area), though woody species were felled closest to the bank in natural/close-to-nature stands. The distance of felling was most likely affected by the distribution of woody plant species, the amount of woody plant biomass potentially usable by beavers, and available species composition. Natural forest stands are not managed, maintained, or harvested, thanks to which willow woody plants naturally occur close to the water (24%), allowing beavers to fulfill their dietary requirements without covering great distances. Previous studies also show that, in general, woody plants are felled near water, and with increasing distance from water, felling decreases, though the range of distances tends to vary at different sites. Some authors [17], for example, reported the most intensive felling 0–10 m from water in a study stretch of 0–40 m, while a study that only examined woody plants up to 10 m from the bank, recorded more than 80% of felling at a distance of 5 m from the water, which is similar to our own results [31]. Elsewhere it was recorded that most felling took place at 11–20 m and less from 0–10 m from water [32]. Similarly, other authors recorded the majority of felled woody plants at distances up to 25 m from the water, with damage to woody plants increasing up to 15 m from water and then decreasing [30]. In a study from Poland [55], for example, authors noted that peak felling distance from water differed in lowland and hilly regions, with the median felling distance in lowlands being 7 m shorter than at higher altitudes.

Overall, forest stand species composition is generally considered the main factor affecting beaver feeding behavior at most sites studied. Beavers are choosy generalists [22], who tend to consume the species dominant in woody availability. However, woody plant distance from water and trunk diameter is also important. This is in contradiction with some results, where trunk diameter was more important than species composition [56], or the selection of woody plants was more pronounced with increasing distance from water [57]. However, the threat of predation [47,52,58] or disturbance [59] may also play an important role, which can significantly change woody plant selection.

Beavers can cause considerable economic losses by felling woody plants in commercial forests [28]. However, population control is problematic in Central Europe as the species is protected as a rare or endangered species [3]. Improved knowledge of beaver feeding behavior, however, could mitigate such damage. Based on our own results, the most threatened commercial woody plant stands are forest monocultures with woody plants up to 20 years old. In line with forest management rules, young plantations damaged by beavers must be replanted; however, these new saplings are also likely to be damaged. To combat this, young plantations must be protected, at great cost, meaning that such sites, particularly those close to water, become unprofitable. We suggest that young stands of valuable commercial species could be protected through the planting of non-commercial woody plant species (willows, poplars, hazel) between the commercial species and the river as a buffer. These are attractive to beavers, particularly as winter food, and grow fast, meaning that stands can be rapidly established at a relatively low cost.

## 5. Conclusions

Forest management has a significant effect on the selection of woody plants in the non-vegetation period. In natural forests, the selection of woody plants by beaver does not change significantly and is comparable to many previous results. However, the natural composition of woody plants was regulated around the rivers in the form of commercial stands with several species of commercial woody plants up to monocultures, where beavers showed a distinct preference for the most available woody plant species, in this case, commercial species such as oaks, leading to economic losses. With different types of management, not only the species selection changes but also the trunk diameter and the distance of felled woody plants from the water, which again affects the amount of damage. To combat this financial loss, we suggest that semi-natural buffer zones of softwood species along watercourses would reduce commercial losses, increase local biodiversity, and restore natural floodplain forests.

## Figures and Tables

**Figure 1 animals-12-02949-f001:**
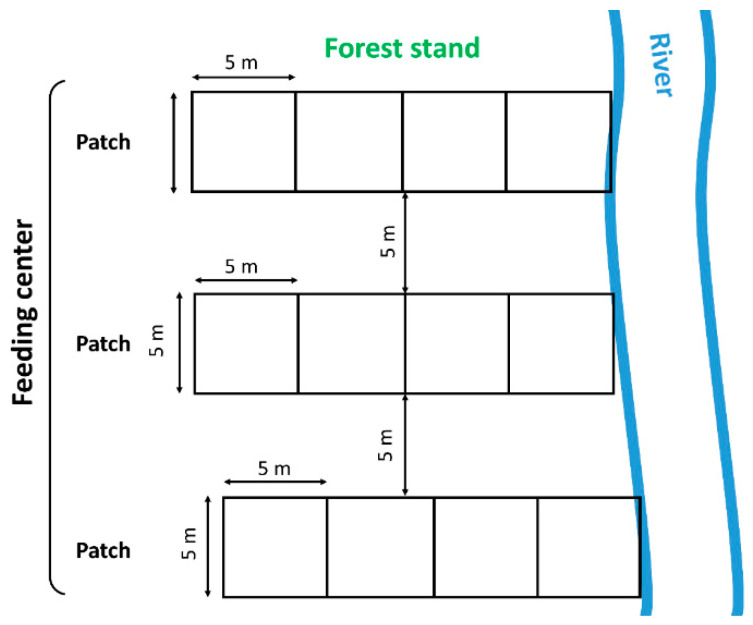
A schematic illustration of the method for staking out study patches in forest stands and their orientation to the watercourse.

**Figure 2 animals-12-02949-f002:**
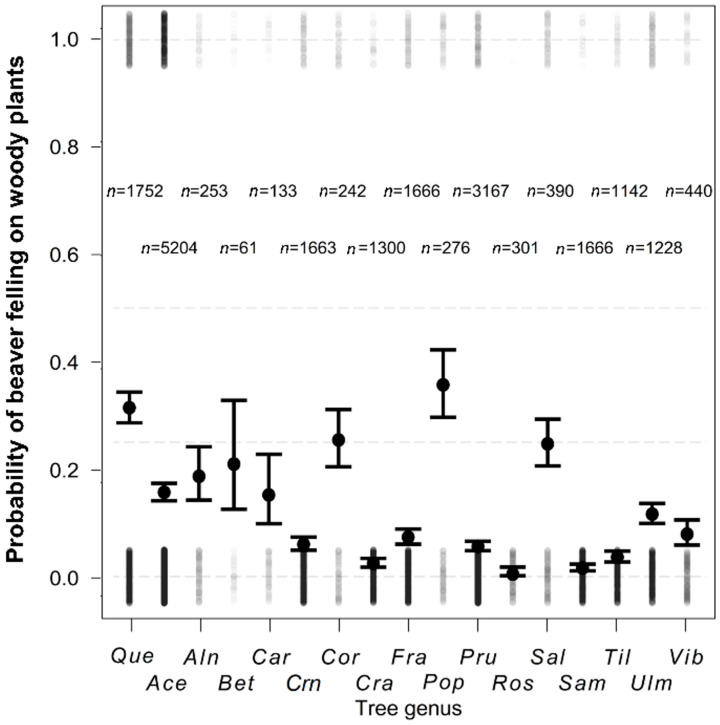
Probability of beaver felling on 17 individual woody plant genera (mean ± SE) after controlling the effect of spatial variance, distance from water, and diameter of trunk. *Que*—*Quercus*, *Ace*—*Acer*, *Aln*—*Alnus*, *Bet*—*Betula*, *Car*—*Carpinus*, *Crn*—*Cornus*, *Cor*—*Corylus*, *Cra*—*Crataegus*, *Fra*—*Fraxinus*, *Pop*—*Populus*, *Pru*—*Prunus*, *Ros*—*Rosa*, *Sal*—*Salix*, *Sam*—*Sambucus*, *Til*—*Tilia*, *Ulm*—*Ulmus*, *Vib*—*Viburnum*.

**Figure 3 animals-12-02949-f003:**
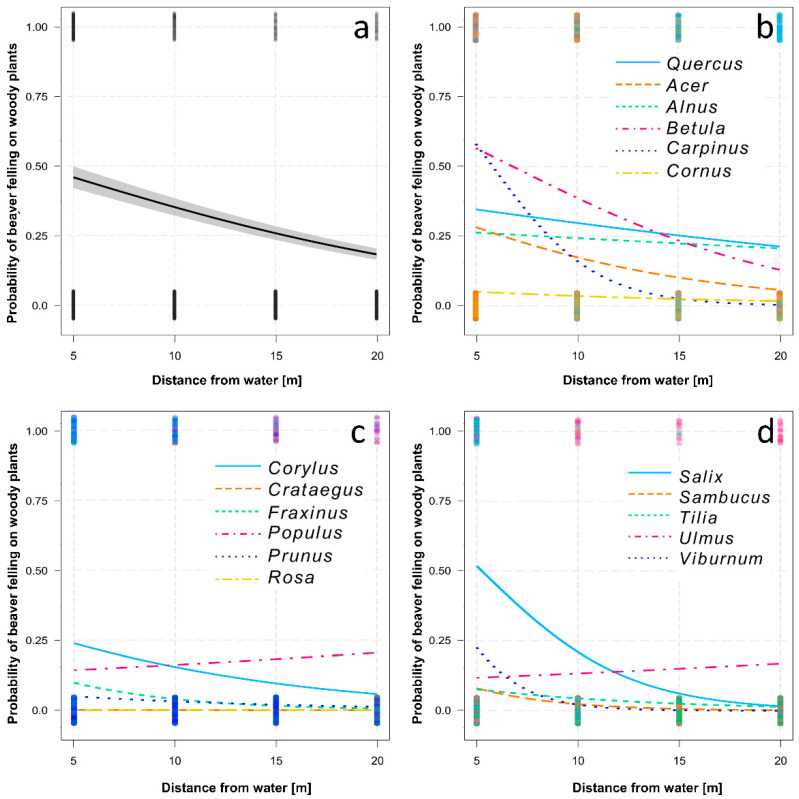
Probability of beaver felling on woody plants with increasing distance from the water: (**a**) overall model after controlling the other effects (spatial variance, diameter), (**b**–**d**) models for individual woody plant genera showing interactions.

**Figure 4 animals-12-02949-f004:**
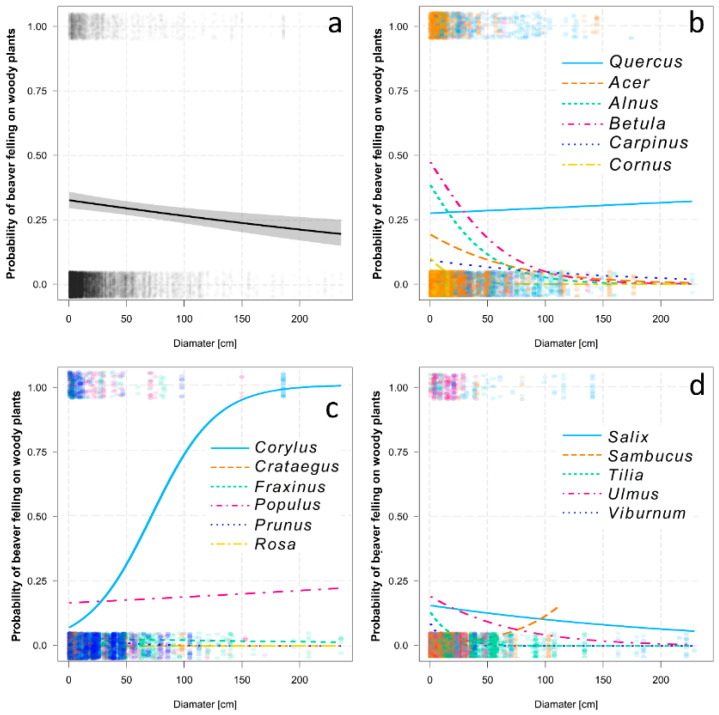
Probability of beaver felling on woody plants with increasing diameter: (**a**) overall model after controlling the other effects (spatial variance, distance from water), (**b**–**d**) models for individual woody plant genera showing interactions.

**Figure 5 animals-12-02949-f005:**
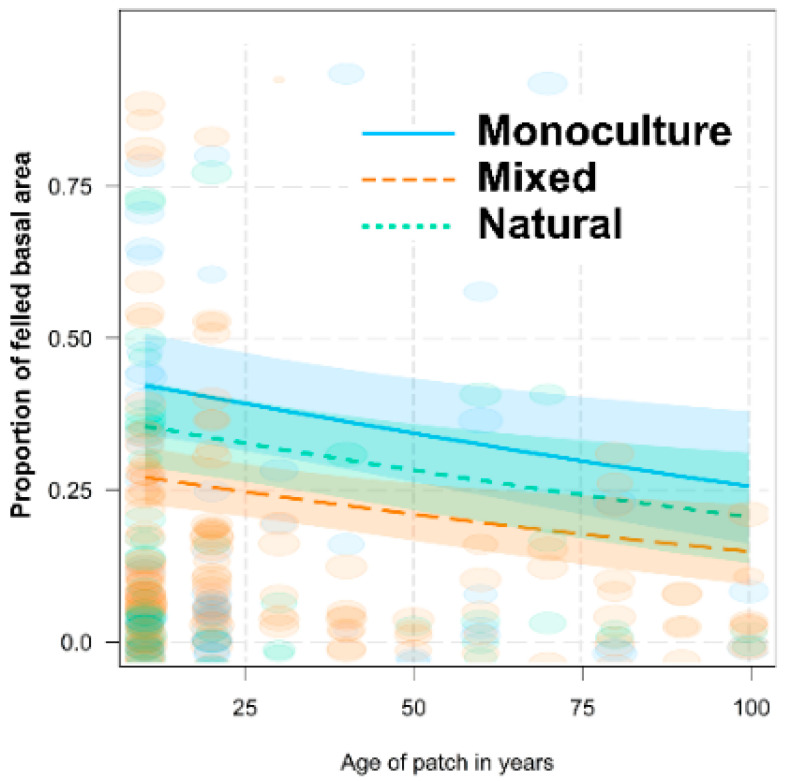
The proportion of felled basal area in a patch in relation to the age of the patch for individual types of forests (estimate ±95% CI). The size of the ellipses reflects the number of woody plants in the patch.

**Figure 6 animals-12-02949-f006:**
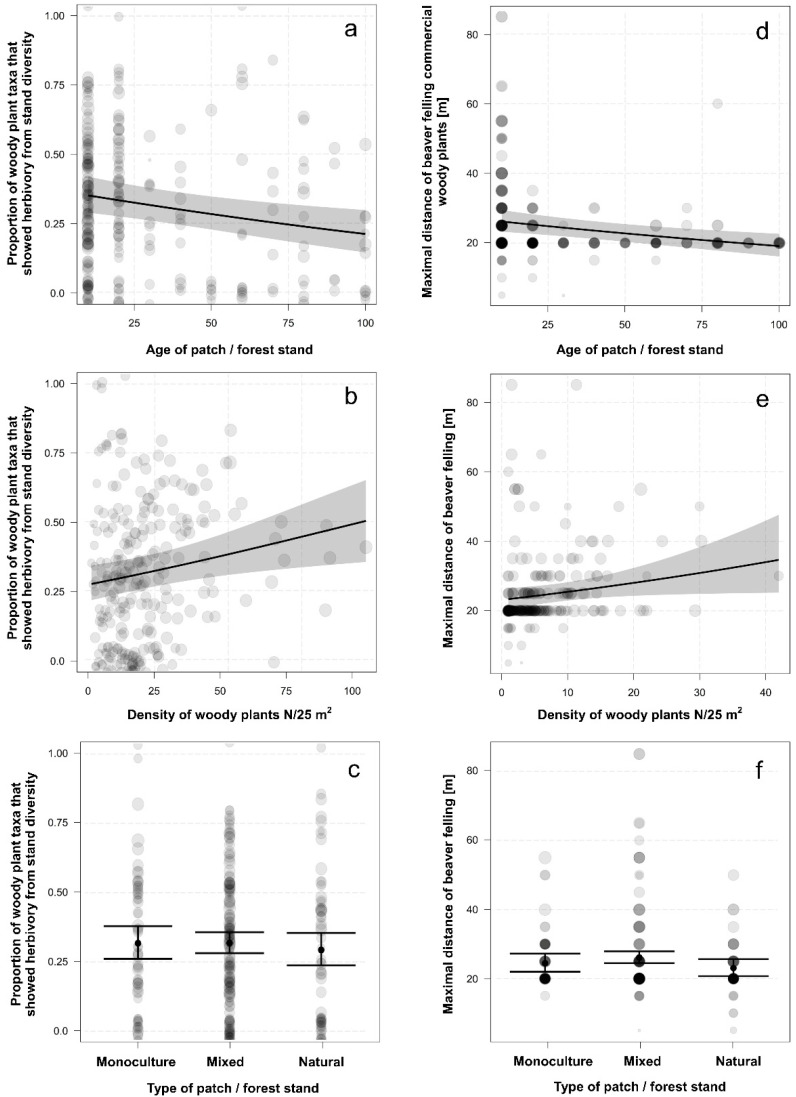
The proportion of woody plant taxa that showed herbivory from patch diversity and the maximal distance of beaver felling in relation to: (**a**,**d**) age of the patch (estimate ± 95% CI), (**b**) the density of woody plants (estimate ±95% CI), (**e**) the density of commercial woody plants (estimate ± 95% CI), and (**c**,**f**) type of the forest patch (mean ± SE). The size of the circles reflects the number of woody plants in the patch.

**Table 1 animals-12-02949-t001:** Availability of woody plants (Available; % basal area) and the proportion felled by beavers (Fell; % basal area) in different patch/forest stands. [comm]—main commercial species.

		Monoculture		Mixed		Natural		Total	
Genus Species	Specific Species	Available	Fell	Available	Fell	Available	Fell	Available	Fell
*Quercus* spp. [comm]	*robur*, *petraea*, *cerris*	**70.63**	**18.31**	**44.05**	**8.27**	6.66	1.03	37.63	8.07
*Fraxinus* spp. [comm]	*excelsior*, *augustifolia*	**25.38**	**2.39**	**15.57**	1.26	3.10	0.24	13.63	1.17
*Populus* spp.	*tremula*, *alba*, *nigra*	0.01	0.00	9.87	**3.72**	**14.28**	**2.17**	9.14	2.41
*Tilia cordata*		0.57	0.00	5.32	0.38	5.49	0.03	4.32	0.18
*Acer campestre*		1.28	0.40	4.52	0.51	4.67	1.22	3.85	0.72
*Ulmus laevis*		0.10	0.00	3.88	0.38	3.99	0.23	3.08	0.25
*Salix* spp.	*alba*, *euxina*, *cinerea*, *caprea*, *triandra*	0.00	0.00	3.40	0.93	**23.71**	**6.58**	9.33	2.57
*Acer pseudoplatanus*		0.00	0.00	2.34	0.42	11.91	3.97	4.97	1.49
*Sambucus nigra*		0.15	0.00	1.88	0.22	1.17	0.00	1.26	0.10
*Carpinus betulus*		0.01	0.00	1.79	0.20	0.93	0.00	1.12	0.09
*Alnus glutinosa*		0.02	0.01	1.54	0.10	11.56	0.92	4.50	0.35
*Crataegus* spp.	*monogyna*, *laevigata*	0.23	0.01	1.20	0.01	0.18	0.01	0.65	0.01
*Prunus avium*		0.26	0.01	1.09	0.06	0.47	0.02	0.70	0.04
*Acer negundo*		0.00	0.00	0.59	0.06	2.18	0.26	0.98	0.11
*Betula pendula*		0.00	0.00	0.56	0.06	2.12	0.38	0.95	0.15
*Prunus padus*		0.89	0.44	0.54	0.00	2.74	0.25	1.34	0.17
*Acer platanoides*		0.09	0.00	0.43	0.10	2.81	1.46	1.14	0.52
*Robinia pseudoacacia*		0.00	0.00	0.41	0.00	0.00	0.00	0.18	0.00
*Viburnum opulus*		0.00	0.00	0.30	0.03	0.11	0.04	0.17	0.03
*Cornus* spp.	*sanguinea*, *mas*	0.25	0.03	0.27	0.04	0.09	0.02	0.21	0.03
*Pinus* spp.	*sylvestris*, *strobus*	0.00	0.00	0.24	0.00	0.16	0.00	0.16	0.00
*Corylus avellana*		0.05	0.00	0.08	0.00	1.01	0.59	0.38	0.19
*Prunus avium*		0.00	0.00	0.07	0.00	0.00	0.00	0.03	0.00
*Euonymus europaeus*		0.00	0.00	0.04	0.00	0.00	0.00	0.02	0.00
*Rosa canina*		0.06	0.00	0.02	0.00	0.01	0.00	0.03	0.00
*Pyrus pyraster*		0.00	0.00	<0.01	0.00	0.00	0.00	<0.00	0.00
*Ligustrum vulgare*		0.00	0.00	<0.01	<0.01	0.00	0.00	<0.01	<0.01
*Rubus idaeus*		0.00	0.00	<0.01	0.00	0.00	0.00	<0.01	0.00
*Glossularia uva crispa*		0.00	0.00	<0.01	0.00	<0.01	0.00	<0.01	0.00
*Aesculus hippocastanum*		0.00	0.00	0.00	0.00	0.63	0.29	0.21	0.10
*Fagus sylvaticus*		<0.01	0.00	0.00	0.00	0.00	0.00	<0.01	0.00
*Malus domestica*		0.00	0.00	0.00	0.00	<0.01	0.00	<0.01	0.00
*Frangula alnus*		0.00	0.00	0.00	0.00	0.01	0.00	<0.01	0.00
*Amorpha fructicosa*		0.00	0.00	0.00	0.00	<0.01	0.00	<0.01	0.00
**Total**		**100.00**	**21.59**	**100.00**	**16.74**	**100.00**	**19.72**	**100.00**	**18.76**

**Table 2 animals-12-02949-t002:** Comparison of individual woody plant genera to *Quercus*, one of the most frequently felled and simultaneously commercial woody plants used as control. Negative t-values mean a less frequently felled wood species, and positive t-values a more frequently felled one. * *p* < 0.05, ** *p* < 0.01, *** *p* < 0.001, . negative t-value.

Genus	Coefficient	SE	t-Value	*p*-Value	Significance
*Acer*	−0.71	0.24	−2.96	0.003	**
*Alnus*	−0.94	0.51	−1.87	0.062	.
*Betula*	1.01	1.02	0.99	0.321	
*Carpinus*	−1.18	0.91	−2.41	0.016	*
*Cornus*	−1.94	0.33	−5.86	<0.001	***
*Corylus*	−0.52	0.31	−1.68	0.092	.
*Crataegus*	−2.29	0.41	−5.53	<0.001	***
*Fraxinus*	−1.69	0.50	−3.39	<0.001	***
*Populus*	1.24	0.48	2.60	0.009	**
*Prunus*	−1.13	0.28	−3.99	<0.001	***
*Rosa*	21.41	7713.78	0.00	0.998	
*Salix*	0.45	0.46	0.99	0.324	
*Sambucus*	−4.85	0.70	−6.98	<0.001	***
*Tilia*	−3.07	0.31	−9.80	<0.001	***
*Ulmus*	−1.46	0.52	−2.83	0.005	**
*Viburnum*	2.93	0.76	3.85	<0.001	***

## Data Availability

Data are contained within the article or Appendix A.

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
