# Peer review of "The Influence of Forestry Management on the Selection of a Non-Vegetative Diet by the Eurasian Beaver (Castor fiber L.)"

_animals, 2022, doi:10.3390/ani12212949_

Round 1

Reviewer 1 Report

Dear Author

The paper is highly interesting because it refers to problematic species in Europe. I have some suggestions that I describe line by line:

Line 36 - instead of 'the preference' and 'the distances' I would use 'a preference' and '- distances'

From line 46 - you should change numers of cited papers. You must start from [1]. In MDPI journals order of citations is important, not alphabetical order. Be careful, because it is a lot things to change.

Don't put a space bteween numbers, eg. [1,2], not [1, 2].

Line 65 - in British English you should write: In winter,  whereas in American it is "In the winter". I suggest British English in whole paper.

Line 79 - sugesstion to discuss with native speaker - This same preference... or The same preference...

Line 88, 325, 328, 330, 331, 332, 350, 352, 354, 355, 357, 364, 365, - modify these sentence, rather don't use Authors surnames, but use passive voice, etc to express what these Authors discovered.

Line 109, 111, 114 - 'to' not 'To'

Table 1, column 3 - put a word 'Available' in smaller size to stay in one line, like others

Line 212 - put a space between the end of a table and first line.

Line 254 - Figure 4 - can you make it a little smaller to stay on previous page without leaving such a big empty space?

Line 265 - Figure 5 - I suggest putting charts in one line, or at least a and b in one line and c in the second one.

Line 276 - Figure 6 - put it in the same way like Figure 5.

Line 320 - autumn

Line 383-386 - these are not conclusions, but results. Please delete it and put something else in the beginning of Conclusions.

From 416 line - References - numbers of pages and issue should not be written in Italics, only name of journals. Whole list of cited papers will be change because of the first remark - modify it according to the order of citation.

Reviewer 2 Report

This is an interesting study focused on assessing foraging behavior by Eurasian Beaver within riparian systems. Overall, I found the study to be novel and a nice addition to existing literature. I do have some comments and critiques which appear below:

Line 47: When you say that Eurasian Beavers were 'exterminated' I think this limits the perception of what really happened. They were extirpated across much of their range but not all. Please revise this text. 

Line 71: Is this to say that other periods are not critical? Consider reforming to 'survive the winter [...'

Line 87: But you did not assess survival. I think this text gives the wrong impression of the study. The difference between occupancy and survival can be subtle but needs to be clarified. Rework so less focused on survival.

Line 215: Perhaps I overlooked it but I didn't see mention of 'control' in the Methods. Please check to ensure methods are properly described.

Lines 216 - 218: Preference is a difficult thing to quantify. Rework as 'most frequently felled"

Fig. 1: What are the vertical bars at the top and bottom of the figure? Are these to designate the different data points in this figure? What do the different colors of these bars represent? Add text as explanation to figure footer.

Line 249: should this be 'basal' area?

Line 258: Rather than 'used' consider 'woody plant taxa that "showed herbivory"' 

There are a lot of figures, some of which do not exhibit what appear to be considerable effect sizes. Consider condensing, or omitting some figures - especially those not showing significant relationships - and instead report effect sizes in the text.

This study was conducted in early spring. Is it clear that all evidence of foraging was associated with the prior winter? Line 283 you draw association with winter diet but is there certainty that all trees felled reflect winter diet?

Line 352: extra dash in this separation

Appendix A1: Are the different colors in the bars of the graph reflective of monoculture, mixed, and natural? If so, say this specifically. As the text in the footer is written it is not clear what you are referring to when you reference these categories.  
